# A New Artificial Intelligence-Based Method for Identifying Mycobacterium Tuberculosis in Ziehl–Neelsen Stain on Tissue

**DOI:** 10.3390/diagnostics12061484

**Published:** 2022-06-17

**Authors:** Sabina Zurac, Cristian Mogodici, Teodor Poncu, Mihai Trăscău, Cristiana Popp, Luciana Nichita, Mirela Cioplea, Bogdan Ceachi, Liana Sticlaru, Alexandra Cioroianu, Mihai Busca, Oana Stefan, Irina Tudor, Andrei Voicu, Daliana Stanescu, Petronel Mustatea, Carmen Dumitru, Alexandra Bastian

**Affiliations:** 1Department of Pathology, Colentina University Hospital, 21 Stefan Cel Mare Str., Sector 2, 020125 Bucharest, Romania; sabina_zurac@yahoo.com (S.Z.); luciana.nichita@umfcd.ro (L.N.); mirelacioplea@yahoo.com (M.C.); liana_ro2004@yahoo.com (L.S.); dragusin_alexandra88@yahoo.com (A.C.); thanatogenesis@gmail.com (M.B.); oana.stefan93@yahoo.com (O.S.); irinafrincu@yahoo.com (I.T.); carmendumitru2004@yahoo.com (C.D.); alexandra.bastian@umfcd.ro (A.B.); 2Zaya Artificial Intelligence, 9A Stefan Cel Mare Str., 077190 Voluntari, Romania; cristian.mogodici@zaya.ai (C.M.); teodor.poncu@zaya.ai (T.P.); bogdan.ceachi@zaya.ai (B.C.); andrei.voicu@zaya.ai (A.V.); daliana.stanescu@zaya.ai (D.S.); petronel.mustatea@umfcd.ro (P.M.); 3Department of Pathology, Faculty of Dental Medicine, University of Medicine and Pharmacy Carol Davila, 37 Dionisie Lupu Str., Sector 1, 020021 Bucharest, Romania; 4Department of Computer Science, Faculty of Automatic Control and Computers, University Politehnica of Bucharest, 313 Splaiul Independenţei, Sector 6, 060042 Bucharest, Romania; 5Department of Surgery, Faculty of Medicine, University of Medicine and Pharmacy Carol Davila, 37 Dionisie Lupu Str., Sector 1, 020021 Bucharest, Romania

**Keywords:** artificial intelligence, tuberculosis, Mycobacterium tuberculosis, Ziehl–Neelsen

## Abstract

Mycobacteria identification is crucial to diagnose tuberculosis. Since the bacillus is very small, finding it in Ziehl–Neelsen (ZN)-stained slides is a long task requiring significant pathologist’s effort. We developed an automated (AI-based) method of identification of mycobacteria. We prepared a training dataset of over 260,000 positive and over 700,000,000 negative patches annotated on scans of 510 whole slide images (WSI) of ZN-stained slides (110 positive and 400 negative). Several image augmentation techniques coupled with different custom computer vision architectures were used. WSIs automatic analysis was followed by a report indicating areas more likely to present mycobacteria. Our model performs AI-based diagnosis (the final decision of the diagnosis of WSI belongs to the pathologist). The results were validated internally on a dataset of 286,000 patches and tested in pathology laboratory settings on 60 ZN slides (23 positive and 37 negative). We compared the pathologists’ results obtained by separately evaluating slides and WSIs with the results given by a pathologist aided by automatic analysis of WSIs. Our architecture showed 0.977 area under the receiver operating characteristic curve. The clinical test presented 98.33% accuracy, 95.65% sensitivity, and 100% specificity for the AI-assisted method, outperforming any other AI-based proposed methods for AFB detection.

## 1. Introduction

Tuberculosis (“consumption”, “phthisis”, or “white plague”) is one of the ancient infectious diseases of humankind. It is produced by Mycobacterium tuberculosis, several other species of mycobacteria are pathogenic in humans (Mycobacterium bovis, Mycobacterium avium intracellulare, Mycobacterium leprae and, in special circumstances, a few others). In 2020, 10 million people had tuberculosis (TB) worldwide, with a yearly death rate of approximately 1.5 million people, thus it is in the top 13 causes of death and second cause of death by an infectious disease after COVID-19 [1].

### 1.1. Rationale for Automatic Detection of Mycobacteria

Diagnosis of TB relies on several methods; mycobacteria identification is one of the most important. In histopathology, identification of mycobacteria requires specific acid-fast stains; the most common one is the Ziehl–Neelsen (ZN) stain, the bacillus appearing red on a blue background. Fluorescent tests such as auramine (golden bacilli on black background) may be also performed but they are more expensive and more difficult to use than ZN [2]. The main problem is that Mycobacterium tuberculosis is a tiny bacillus (length 2–4/width 0.2–0.5 microns) and it must be searched for in a 2 × 3 cm fragment of tissue (6,000,000,000,000 square microns) by thoroughly examining hundreds or thousands of microscopic fields 0.5 mm in diameter.

A pathologist will experience fatigue, diminished attention, and may end by postponing examination. In fact, a pathologist will not examine the whole slide but those areas with lesions more suspicious to present bacilli (necrotic areas and epithelioid granulomatous inflammatory infiltrate with or without Langhans multinucleated giant cells), in order to reduce the time of examination. Attempts of automatic detection of mycobacteria represent the logical answer to this problem.

### 1.2. Literature Review

The first method of artificial intelligence (AI) detection of AFB was developed by Veropoulos et al., in 1999 on smears stained with auramine [3]. Several other studies proposing methods of automated detection of AFB on ZN stains evaluated smears. Other than delCarpio et al., and Law et al. (who evaluated scans of slides containing the whole section present on the slide whole slide images (WSIs)) [4,5], all the other studies evaluated images captured with cameras (small parts of slides) [6,7,8,9,10,11,12,13,14]. The specificity and sensibility varied from study to study as shown in Table 1.

There are other studies published on automatic AFB detection on tissue using WSIs [15,16,17,18,19]. Analyzing tissue is more difficult than analyzing smears. No matter how small a bacillus is, by the mere sectioning of paraffin blocks the bacillus will be cut in different incidences, in various relationships with the adjacent cells/structures. Additionally, artifacts created by sectioning are more complex with a special emphasis on conglomerated red blood cells—the membranes of adjacent red blood cells compressed one on top of the other is a very close mimic of AFB in a ZN stain.

Xiong et al. (2018) developed a convolutional neural network (CNN) model pretrained on the CIFAR-10 dataset [15]. They used a training set of 45 slides (30 positive and 15 negative) digitalized as WSIs with a KF-PRO-005 Digital Section Scanner (Ningbo Jiangfeng Bio-information Technology Co., Ltd., Ningbo, China). Annotations were made with ASAP software, the dataset consisting of 96,530 positive and 2,510,307 negative 32 × 32 pixels patches. Several augmentation techniques were used, extending the positive dataset to 578,191 patches. The test set consisted of 201 slides (108 positive and 93 negative). The test slides were divided into 32 × 32 pixels patches that were then fed to the algorithm. The model analyzed patches from slides and labeled them as positive when the probability score was over 0.5. Only one positive patch is necessary to label the entire WSI as positive. Xiong’s et al., method of diagnosis is completely automated—the classification of WSIs is performed by the algorithm and does not involve a human examiner. The test was performed twice. After the first run, the false positive and false negative cases (labeled as such based on human evaluation) were reevaluated by two pathologists; seven cases were primarily missed by pathologists and six cases were not suitable for analysis due to the poor quality of the scans. In the end, the performance metrics of the model are 97.94% for sensitivity and 83.65% for specificity. Based on the data available in their paper, the accuracy of Xiong et al.’s model is 90.55%. The model has a very good sensitivity, catching most of the bacilli but the specificity is too low to give many false positive results. The dataset includes a relatively small number of cases possibly restricting the color variability of the input space that is modeled.

Yang et al. (2020) [16] constructed a pipeline that consists of combining a CNN model (Inception-V3) for tile-based classification and a logistic regression (LR) model for WSI classification. The CNN model was trained to identify tiles (patches) with AFB initially using a dataset of patches of 256 × 256 pixels originating from 14 WSIs (6 positive and 8 negative slides digitized with an Aperio AT Turbo scanner (LeicaBiosystems, Vista, CA, USA)). Then, the model was retrained using a semi-supervised active learning framework that employed the initial dataset completed with new patches originating from 19 negative WSIs. The models were validated on a separate validation set of patches with F1 scores of 99.03% and 98.75%. Then, the retrained CNN model was used to classify patches (in positive and negative) from a set of 134 WSIs (46 positive and 88 negative), the results being used by the LR model to classify the digitized slides. Yang et al., developed an AI-assisted diagnostic method. Their pipeline of analysis creating a score heatmap of AFB probability tiles overlaid onto the WSI. The pathologist examines the areas within the heatmap and confirms the positivity of the WSI. The WSI-level metrics of the pipeline were above 80%: sensitivity 87.13%, specificity 87.62%, and F1 80.18%. The analysis produces a score heat map overlaid on the WSI, guiding the pathologist in the analysis of the probable positive tiles. However, the low specificity (87.62%) indicates that the model identifies numerous patches as false negatives and forces the pathologist to examine thousands of patches suggested as positive. The model yields approximately 4,500 tiles false positive in a 1 × 1 cm^2^ section of tissue. In the end, the time and energy spent analyzing the results might end up more than in the classical (“manual”) microscopic examination. Additionally, the diversity of the dataset is limited. The patches are selected from 14 cases (only 6 positive) with a further addition of 19 negative ones.

Lo et al. (2020) [17] developed a CNN model to detect mycobacteria based on a dataset of 1815 patches (blocks) of 20 × 20 pixels (613 positive and 1202 negative) out of which 80% were randomly selected for training, the remaining 20% being kept for validation. Additionally, another 1383 negative patches mimicking AFB (mast cells, background stain, etc.) were selected. The annotations were performed on nine positive slides digitized with the help of a ScanScope XT whole-slide scanner (Aperio, Vista, CA, USA). The model used in the process was a pretrained CNN—AlexNet [20] of five convolution layers. The final three layers were fine-tuned for the target tasks and the blocks from the dataset were resized to 227 × 227 pixels to match the AlexNet architecture. The level of cut-off was established at 0.5. The performance metrics of Lo et al.’s model were 95.3% accuracy, 93.5% sensitivity, and 96.3% specificity. The dataset is significantly smaller than the one we used and was extracted from only nine slides. The results on the validation set are reported at patch level; no WSI analysis is provided.

Pantanowitz et al. (2021) [18] developed an algorithm based on a dataset created from 441 slides scanned with two types of scanners: Aperio AT2 (Leica Biosystems) and Hamamatsu Nanozoomer XR. The dataset included 1,117,586 patches (5678 positive and 1,111,918 negative) selected from 441 WSIs (62 positive and 379 negative) and was separated in three groups: the dataset used for training consisting of 1,054,395 patches (4629 positive and 1,049,766 negative) selected from 418 slides (47 positive and 371 negative); the dataset used for analytical validation (40,957 patches (449 positive and 40,508 negative) selected from 12 WSIs (9 positive and 3 negative)); and the dataset used for testing (22,244 patches (600 positive and 21,644 negative) selected from 11 WSIs (6 positive and 5 negative)). The annotations were made using the aetherSlide application. Two deep CNNs were used in the process, one with high sensitivity and the other with high specificity. The model with the highest accuracy (0.960 at the image patch level—calculated as area under the receiver operating characteristic (ROC) curve (AUC)) in the validation test was selected and used in further clinical validation. Pantanowitz et al., developed an AI-assisted screening method. Their tool displays a gallery of patches with their corresponding probability scores and the WSI to give the possibility of examining the suspicious patches in context. The clinical validation was performed on 138 slides. It consisted of a blind evaluation performed by two pathologists with different levels of expertise by classical “manual” microscopic evaluation of the slides, evaluation of the WSIs, and algorithm-assisted evaluation versus a gold standard represented by the signed-out assessment. The performance metrics of Pantanowitz et al.’s model were 84.6% accuracy, 64.8% sensitivity, and 95.1% specificity.

Zaizen et al. (2022) [19] developed an algorithm to detect AFB using a pre-trained HALO AI CNN. The dataset consisted of 506 AFB annotated on two autopsy cases with TB; the negative ones including two types of artifacts (nuclei of type I epithelial cells as well as fibrin and hyaline membrane) originating from 40 negative biopsies. The slides were digitized using a Motic EasyScan scanner (Motic, Hong Kong, China) and the annotations were performed using the HALO platform (version 3.0; Indica Lab, Albuquerque, NM, USA). Zaizen et al., also developed an AI-supported diagnosing method. Each patch identified as probably positive by the algorithm was evaluated by six pathologists by consensus. The clinical test included 42 cases, the 16 positive ones were either patients diagnosed with mycobacteriosis by bacteriological tests performed on material harvested during bronchoscopy, or patients who developed mycobacteriosis during the follow-up. The performance metrics of Zaizen et al.’s model were 86% sensitivity and 100% specificity.

There is another study presenting an algorithm of automated AFB detection on tissue, but it was developed on pictures (24-bit RGB images at a resolution of 800 × 600 pixels acquired using a digital camera) and not on WSIs. The accuracy obtained was 77.25% [21].

### 1.3. Novelty of Our Method

We propose an automatic method of identifying AFB using deep neural networks. These will be trained to process WSIs and indicate the AFB location. The pathologist analyzes the patches suggested as positive and decides if the slide is positive or not (AI-assisted diagnosis).

Our algorithm has several advantages compared to previous works. Our dataset is much larger, more diverse, and more carefully selected than the other datasets.

#### 1.3.1. Dimension

Its positive component is almost 3 times bigger than the next largest one (263,000 positive patches in ours vs. 96,530 in Xiong et al.’s dataset [15]), which is 429 times bigger than the smallest ones (506 in Zaizen et al.’s set [20]). Our negative patches are 7 times more numerous than the second largest one of Pantanowitz et al. [18] (7,000,000 vs. 1,111,918). Further applied augmentation techniques (both as position—rotations, shifts, crops, etc.—and in image properties—brightness, contrast, saturation, etc.) expanded our positive group of patches to more than 2,500,000.

#### 1.3.2. Diversity

Our dataset resulted from annotation of a total of 510 WSIs; 110 WSIs were positive. The other datasets were constructed based on 2 up to 47 positive WSIs, the variability of the positive images being much lower. Additional consideration of the variability in tinctoriality of ZN staining shows that the diversity of our dataset is significantly increased. We included bacilli in more numerous and diverse backgrounds and in a greater variety of ZN stains.

#### 1.3.3. Model and Augmentations

We used a large set of composable augmentations from which we generated both hard-labeled and soft-labeled training samples. We proposed specific modifications to the original RegNet-X architecture that was adapted such that it better models the domain task.

#### 1.3.4. Case Selection

Our dataset was built after a strict confirmation of positive cases. Additionally, to prevent the situations when the human examiner is not able to identify few bacilli in a paucibacillar TB, cases with specific TB morphology but with negative ZN stains were excluded. One author had to reclassify several cases [15] and another reported a very high number of cases identified as positive after AI-assisted examination (seven newly identified cases out of a total of nine positive ones) [19].

## 2. Materials and Methods

We started by selecting ZN-stained slides originating from positive and negative cases. (Section 2.1). The ZN-stained slides were scanned and annotated, in the end more than 260,000 positive and more than 7,000,000 negative patches of 64 × 64 pixels were selected. (Section 2.2). The dataset was further expanded by different augmentation techniques. (Section 2.3). We identified and customized a deep learning architecture suitable for our task. (Section 2.4). The model was validated on a validation dataset consisting of 286,000 patches (validation set) different from the dataset used for training. (Section 2.5). The model configuration with the best results in validation was further tested in clinical trials. (Section 2.6). In this phase, the scanned image of the ZN-stained slide was uploaded on a platform, divided in 64 × 64 pixels patches, and each patch was analyzed by the algorithm. The algorithm returned a score for each patch and the pathologist received a list of patches sorted in descending order by their corresponding score. The pathologist analyzed the patches, both separately and in context on the slide and decided if the patch was positive or not. Based on this evaluation, the pathologist decided if the slide was positive (one single positive patch is sufficient to diagnose the slide as positive) or not (AI-assisted diagnosis).

The pipeline’s performance was measured twice:-First evaluation (“validation”) was performed using patches from pre-selected regions on the slides. On the one hand, the validation gave us the possibility to evaluate the performance of several architectures and allowed us to choose the best model for further use. On the other hand, the results obtained on other pre-selected patches (from slide areas used for active learning) were analyzed one by one by pathologists in order to establish errors (positive patches labeled as negative and vice versa; negative ones falsely labeled as positive). The mislabeled patches were correctly re-labeled and used for re-training and finetuning the model, thus improving its performance.-Second evaluation (“clinical testing”) was performed using WSIs. Each WSI was segmented in 64 × 64 pixels patches and all patches were fed to our model for analysis. The model examined each patch and gave a score of probability (0 to 1)—the probability of the patch to belong to the positive group of patches used for training = to present mycobacteria. The results were displayed as a column of patches with their class score, arranged in descending order of the score (i.e., the patch with the maximum score was listed first). A threshold must be established for a patch to be considered positive; the obvious choice should be 0.5. However, in our testing data we noticed that almost all patches with scores between 0.5 and 0.7 were negative. Our model was not classifying the WSI as positive or negative, instead it revealed to the pathologist the patches that are more probable to harbor bacilli, leaving the final decision of the diagnosis of WSI to the human examiner. (This is considered AI-assisted diagnosis).

### 2.1. Case Selection

We analyzed the archives of the Department of Pathology of Colentina University Hospital from 2010 to 2022. We selected 2187 cases with ZN stains mentioned in the histopathology report. Consultation cases were excluded.

All the cases were re-evaluated both on H&E and ZN stains available in the archive by SZ (a senior pathologist with 23 years of expertise). Cases with discordances between the initial histopathological report and SZ’s re-evaluation were excluded.

-Positive cases group: cases reported as diagnosed with TB with ZN-stain slides positive and reconfirmed as such by microscopic reevaluation.-Negative cases group: cases without AFB bacilli in ZN stain (both primary—at the moment of diagnosis) and confirmed diagnosis of other illnesses than tuberculosis. Cases with histopathologic appearance highly suggestive of tuberculosis (epithelioid granulomatous inflammatory infiltrate with multinucleated giant cells and coagulative necrosis conserving reticulin network in Gömöri stain—specific morphological aspect of caseating necrosis) and negative ZN stain were NOT included.

All the cases were tissue fragments (either biopsies or surgical specimens) received by our department as fresh or formalin-fixed tissue. After the macroscopic examination (grossing), the fragments were immersed in 10% buffered formalin until the next day (18–24 h), routinely processed to paraffin (automatic tissue processors Leica ASP 200S (see Appendix A) and Leica Peloris 3 (see Appendix A) were used), embedded in paraffin blocks (embedding stations ThermoFisher Microm EC 1150 H, Leica EG 1150H, Sakura Tissue Tek and Leica Arcadia), sectioned at 3 microns thick (semi-automated Rotary Microtome Leica RM2255 and RM2265) and stained with ZN staining kit (Ziehl–Neelsen for mycobacteria—microbiology, BioOptica Italy) (see Appendix A).

H&E slides were used only for analyzing the morphologic lesions, to confirm the diagnosis in positive cases, and to exclude from the negative group the cases with high-morphology suggestive of TB but without AFB in the ZN stain. For our study we use only the ZN-stained slides classified as positive and negative as previously described.

ZN-stained slides were scanned using both manual and automatic scanners, each slide being entirely scanned as whole slide image (WSI) in “.svs” format. The manual scanner was provided by Microvisioneer, Esslingen am Neckar, Germany, and consisted of a Camera Basler Ace 3.2 MP (acA2040-55uc) with Sony IMX265 Sensor and Microvisioneer manual WSI Software Professional Edition. The automatic scanner was a Leica Aperio GT450.

Finally, we obtained 570 WSIs: 133 positive and 437 negative; 510 WSIs—group A (110 positive WSIs and 400 negative WSIs)—were used for training purposes while the remaining 60 WSIs—group B (23 positive WSIs and 37 negative WSIs)—were used for testing (Table 2).

### 2.2. Annotation Process

The WSIs from group A were annotated by 7 pathologists with various experience (Appendix A) using an in-house platform for annotation and Cytomine application (Cytomine Corporation SA, Liège). Positive areas were identified either as patches (less than 64 × 64 px) in our in-house annotation platform or point-like annotations of the bacillus in Cytomine platform. Negative samples were drawn either from WSIs labeled as negative or from manually annotated negative areas inside WSIs labeled as positive. Patches selection from negative WSIs was performed in two steps: firstly, the WSI area was filtered to contain a sufficient amount of tissue (versus background); secondly, a 64 × 64 patch was sampled via a uniform distribution from this area. In the end, we obtained a pool of negative samples containing more than 700 million patches before applying any augmentations. Examples of positive and negative areas are depicted in Appendix A.

### 2.3. Image Augmentation Techniques

Even though the dataset obtained via the annotation process contains more than 260,000 positive examples, there is a large diversity that the staining process induces to the color space of both positive and negative WSIs. In order to mitigate this, we have employed extensive augmentation techniques to cover a wider variety in WSIs. The augmentation transformations were applied to all training patches. These included random rotations in the range of 0 to 90 degrees (clockwise and counterclockwise); random shifts; random crops; and random brightness, contrast and saturation changes. In addition to these specific transformations, we also extracted positive patches around the annotated AFBs by shifting a maximum of 24 pixels in any of the two axes. Since all transformations were applied in a chain, specific to sample interpolation techniques, we considered that training examples have diversified by at least one order of magnitude.

### 2.4. Deep Learning Model Development and Training

Our patch-based classifier for AFB detection is based on RegNetX4 architecture. This deep convolutional neural network manages to yield state-of-the-art performance while preserving simplicity and speed. It has the advantage of requiring less hyperparameter tuning, which is an important consideration when dealing with the large amount of data and data manipulation techniques used in our setting. In order to better fit the task at hand, we have adapted the architecture through various custom modifications:We reduced the kernel size in the stem layer (plausible morphology of bacillus can be constrained in a 3 × 3 convolution filter or 5 × 5 convolution filter);We reduced the number of strided convolutions, as an overall larger receptive field in the final stages is not necessarily helpful due to the low spatial size of the target class;We employed parallel dilated convolutions (i.e., selective kernel convolutions [22], atrous convolutions [23]) in order to accommodate morphologies that are not necessarily captured in a 3 × 3 filter while still keeping a reasonable amount of trainable parameters;We opted for reflection padding instead of zero padding in all padded convolution layers to reduce locality bias learned by the network in order to be more robust to bacillus positioning inside boxes served at inference or testing time.

The network variant we used contains less than 160 million learnable parameters, allowing for adequate inference speed even when not using high-end hardware, without any degradation of the performance metrics.

We trained our model in a distributed fashion using parameter replicas for each Graphics Processing Unit (GPU) and gradient averaging before broadcasting parameter updates. We used a batch size of about 2048 per GPU with positive and negative examples roughly evenly split in order to mitigate the severe class imbalance. We experimented with various proportions (positive vs. negative) starting from 25–75% up to 75–25% using 5% increments. We limited our learning procedure to a maximum of 100 million samples seen (including augmented patches). The optimizer used was AdaBound where the step size α is provided by a linear warm-up cosine scheduler with periodic restarts [24]. Inference is performed only on the parts of the WSI containing a relevant amount of tissue. Filtering the WSI was performed using the same method as for filtering the training areas used for extracting patches. Depending on the WSI size and CPU threads used for WSI patch area extraction our baseline processing pipeline (i.e., 1 CPU thread) manages to process a WSI in 5 to 15 min.

### 2.5. Model Validation

We constructed a set of validation patches by annotating several areas collected from 37 WSIs; obviously, none of the areas selected for validation were previously used for training. The areas thus collected have been divided in non-overlapping patches of 64 × 64 pixels, which were subsequently annotated by the team of pathologists as either positive or negative. We have obtained 286,000 validation patches of which 15,000 are positive and 271,000 negatives. The class imbalance is intentional, as it is much more likely for the model to generate false positives than false negatives and this data distribution is much closer to real conditions than a balanced one.

### 2.6. Testing Process

For the testing process four teams of two pathologists each were involved; each team included pathologists of similar experience (Appendix A). We compared three types of results: B1—results obtained by examining slides with a bright-field light microscope; B2—results obtained by examining the WSIs scanned with a Leica Aperio GT450 automated scanner; and B3—results obtained by algorithm-aided WSIs evaluation. A wash-out period of 2 weeks between each type of evaluation was respected. For each case, in each scenario (B1, B2, and B3), the pathologist registered the status (positive or negative) and the time required to reach the diagnosis. No time limit was established for examination of either slides or WSIs. All the results were compared with the “gold standard”—the original histopathological report reconfirmed by H&E and ZN stains reexamination (see Section 2.1).

### 2.7. Statistical Analysis

Model validation was performed using the validation set that the team of pathologists had produced. A receiver operating characteristic (ROC) curve was plotted to describe the diagnostic ability of the model to classify patches as containing AFB or not. The imbalance we have imposed between the number of positive and negative patches in the validation dataset has led us to select the Precision–Recall curve as a useful measure to compute the area under the receiver operating characteristic curve (AUC) for (AUPR). Due to the same reason, we also computed the F1 score and Matthew’s correlation coefficient (MCC). All metrics were computed using Python libraries scikit-learn and matplotlib.

The performance metrics used to evaluate our proposed method in clinical tests are listed in Appendix A.

Statistical significance of the difference between two groups was analyzed using the χ2 test, where applicable. Statistical significance was defined as *p* < 0.05, and all statistical analyses were performed using the EXCEL program.

## 3. Results

### 3.1. Internal Validation

We have evaluated the model on the validation set. Accurately validating on entire WSIs would require slides that are completely annotated (to have each AFB indicated by an expert). The cost of obtaining such data is prohibitive and requires compromising on staining diversity, tissue morphology, and artifact types, as opposed to performing validation on selected interest areas from multiple WSI.

The evaluation of the configurations has been performed on the best checkpoint identified during the training process. We have obtained an AUC for the ROC curve (Figure 1) of 0.977 and an AUPR for the Precision-Recall curve of 0.843 (Figure 2). The sensitivity, specificity, F1-score and MCC curves are described in Figure 2. It is important to note that the AUC value the model obtained for the ROC curve is considered to be excellent for models used in medical testing [25,26,27]. By setting an arbitrary value of 0.5 for the decision threshold during validation (patches over the threshold are considered positive while under the threshold they are negative) we obtained an accuracy of 0.969, a sensitivity of 0.877, a specificity of 0.974, an F1-score of 0.923, and MCC of 0.745 (Figure 3).

Most frequent false positive findings during validation (at patch level) were given by red blood cells, mast cells, and fibrotic septa. Parts of red blood cells mimic AFB (Figure 4). In fact, even in classical microscopic examination of ZN stained slides, the pathologist has difficulties in differentiating between bacilli and the periphery of red blood cells, especially in case of congestion (when several red blood cells are compressed within a narrow capillary). In addition, since ZN stain has quite important variation, the level of acid–alcohol discoloration can be low, thus preserving a more intense coloration of red blood cells (red or bright pink instead of pale pink)—the internal control for a proper ZN stain is a pale-pink color of red blood cells.

Granules from the cytoplasm of mast cells are colored in purple in ZN by methylene blue that is used for staining the background (see Appendix A for the protocol of ZN stain). Additionally, parts of a mast cell cytoplasm can be confused for an AFB when the patch includes a very small part of the cell (Figure 5).

All patches with scores over 0.7—either positive or negative—were re-evaluated by pathologists. This analysis revealed that very few patches with a negative score over 0.7 (i.e., “more likely similar with negative training data set”) were a false negative. Several patches with a positive score over 0.7 (i.e., “more likely similar with positive training data set”) were erroneously labeled as such. We employed an active learning strategy for training and fine-tuning the model. To this end, we selected a fine-tuning holdout set consisting of several areas from the training WSIs that were non-overlapping with the annotations (either positive or negative). First, the model was trained from scratch with the available data. Then, inference was performed on the validation set, and the validation metrics were assessed (e.g., sensitivity, specificity, F1-score, etc.). The model then was used to classify the areas in the holdout set (which contain both positive and negative patches). The results obtained for the holdout set classification were analyzed by the pathologist team, and the mislabeled patches are correctly relabeled as negative or positive. In the end, the model was retrained for fine-tuning with these new patches. Performance was further improved by performing several iterations of this active learning cycle for the data. Given this process, as described in Figure 6, the model can be easily adapted to new conditions and variations when we obtain new WSIs.

### 3.2. Clinical Testing

Our test group included 37 males (61.67%) and 23 females (38.33%) with a median age of 42.25 years (the youngest patient was 1 year old and the oldest was 80 years old). The specimens were represented by: lymph nodes—35 cases (58.33%), lung—10 cases (16.67%), skin—7 cases (11.67%), striated muscle—5 cases (8.33%), and intestine—3 cases (5.00%). A total of 23 cases were diagnosed as tuberculosis (38.33%) while 37 cases (61.67%) were inflammation other than tuberculosis or cancer (Figure 7): cat scratch disease 5.00%, sarcoidosis 6.67%, unspecific granulomatous inflammation 8.33%, Kikuchi 3.33%, toxoplasmosis 1.67%, unspecific inflammation 8.33%, reactive lymphadenitis 10.00%, non-Hodgkin’s lymphoma 6.67%, Hodgkin’s lymphoma 6.67%, and carcinoma 5.00%. All the cases of tuberculosis had AFB present in ZN stain; obviously, no AFB were present on ZN stain in the other cases. In order to avoid a possible bias in evaluating AFB presence due to correct identification of the lesion (i.e., diagnosing other disease than TB based on morphology alone), all the negative cases were selected to present either necrotizing granulomatous inflammation (cat scratch disease or unspecific granulomatous inflammation), granulomas (sarcoidosis), epithelioid histiocytes (toxoplasmosis), necrosis (Kikuchi, unspecific inflammation, lymphomas, or carcinomas), or florid histiocytosis (reactive lymphadenitis, or unspecific inflammation). No clinical data were available to the pathologists when examining either slides or WSIs.

#### 3.2.1. WSIs Analysis

The results of analyzing WSIs by pathologists showed interesting results (Table 3). Accuracy (capacity to identify closer to the true value) was, with one exception, higher than 0.8333 (varied from 0.6167 to 0.9333). Sensitivity (capacity to identify true positives) varied from 0.3913 to 0.9565 and specificity (capacity to identify true negatives) varied from 0.7567 to 0.9459.

The accuracy (*p* = 0.1), precision (*p* = 0.09), and specificity (*p* = 0.06) had a general tendency to increase as the experience of the pathologist increases, but there was no uniformity towards an increase in the sensitivity with experience (*p* = 0.25) (Figure 8a–d).

When looking at sensitivity, specificity, precision, and accuracy in correlation with experience in analyzing WSI (exposure to WSI) we identify a statistically significant association for specificity (P chi test 0.004), precision (P chi test 0.008), and accuracy (P chi test 0.012), but not for sensitivity (P chi test 0.06) (Figure 9a–d).

#### 3.2.2. Slide Analysis

Slide analysis (microscopic examination) revealed much better results than those obtained on WSIs (Table 4). The senior pathologists had one to three errors per person (senior pathologist 1A—2 errors, senior pathologist 1B—3 errors, senior pathologist 2A—3 errors, senior pathologist 2B—one error; all but one error were false negative); the pathologists had more errors—pathologist A—12 errors, pathologist B—4 errors; 4 of them were false positive and 12 false negative) while residents had 29 errors (resident A 12 errors, all false negative, resident B—17 errors, 2 false positive, 15 false negative). The results were much better than those obtained by evaluating WSIs but algorithm-assisted evaluation had better results than human evaluation either on WSIs or slides. In fact, our model results (AI-assisted evaluation) were better or similar to pathologists examining slides. Senior pathologist 2B was identical, with only one false negative result, for the same case. The resulting accuracy for our model was 98.33% with only one false negative result—sensibility of 95.65% and no false positives—specificity of 100%.

#### 3.2.3. Time Analysis

Time dedicated for WSI examination varied from 10 s to 80 min with an average time of 11.43 min per WSI. The average time of examination varied between examiners from 5.48 min to 17.06 min with shorter times for positive slides and longer for negative ones (either true or false negatives). In fact, for every pathologist, the longest time of examination was recorded for negative cases (true negative for seven examiners and false negative for the remaining one) and the shortest for true positive ones (Table 5). No relation with experience or prior exposure to WSI was identified.

When examining slides, the pathologists spent less time than for WSIs. The overall interval varied from 3 s to 49 min with an average of 5.25 min (Table 6).

Time used by pathologists in AI-assisted examination varied from 9 s to 2.002 min for positive slides (average 0.61 min). In most of the cases, the AFB were present in the first or the second patch in the list. In one case the pathologist examined 25 patches to find a convincing AFB.

We exemplify two cases that required pathologists 32–33 min for examination (an average of 12–13 min for classic microscopic examination) (Figure 10 and Figure 11). In negative cases, a maximum of 4–5 min weas necessary for confirmation of negativity. Average time needed for AI-assisted examination was 1.85 (1 min 51 s).

#### 3.2.4. Error Analysis

We analyzed the errors made by pathologists when evaluating WSIs. To our surprise, human examination of WSIs results in an amazing proportion of 31 WSIs of a total of 60 cases that were erroneously interpreted (51.67%) with a total of 71 misinterpretations of 480 evaluations (Table 7, Figure 12a). Even when residents were excluded, errors occurred for 21 WSIs (35%)—more than one third of the cases (Appendix A, Figure 12b).

## 4. Discussion

Diagnosis of TB can be difficult. A complex interpretation of clinical and radiological images supported by immunological, bacteriological, histopathological, and molecular tests is needed. Paucibacillary lesions are particularly difficult to diagnose. Sputum and/or tissue examination often fail to identify AFB. Bacteriological tests are more successful in identifying mycobacteria than pathology (up to 50–80% more sensitivity for bacteriology compared with histopathology) [28] but the main drawback of the method is the time required by cultures—average of 14–21 days but it is not unusual to take up to 6–8 weeks [29]. PCR and bacteriological tests may also offer divergent results [30]. Immunohistochemistry for mycobacteria is expensive and due to the small dimensions of the bacillus, can be difficult to interpret in paucibacillary lesions.

Histopathologically identification of AFB in the appropriate morphological milieu represents the most precise diagnosis of TB possible because it corroborates the presence of specific lesions with the presence of the bacteria. TB is a form of “specific chronic inflammation”, i.e., inflammation with microscopic lesions so characteristic that, by their presence alone, one can affirm with certitude that the culprit provoking the morphologic alterations is a species of Mycobacterium. The lesions consists of confluent epithelioid granulomas with centrally located Langhans multinucleated giant cells and caseating necrosis. In these cases, the diagnosis requires only a routine H&E stain. However, in different circumstances (early lesions, associated illnesses such as cancers, immunosuppression or (auto)immune diseases, simultaneous infection with other microorganisms, etc.), this typical morphological picture is altered and several special stains are needed for diagnosis: Gömöri staining for reticulin and van Gieson Weighert for elastic fibers (to prove the preservation of reticulin and elastic fibers in necrotic area); ZN or auramine (to identify AFBs); some other special stains (Giemsa, Gram, Grochott, Warthin Starry, etc.) to exclude the presence of other microorganisms; in some cases immunohistochemical tests for mycobacteria; and/or polymerase chain reaction (PCR) for Mycobacterium tuberculosis are performed. Moreover, clinical, blood tests (QuantiFERON-TB), and imaging data are corroborated in order to establish a diagnosis of TB [31,32].

Understanding the details of the histopathologic diagnosis of TB is mandatory in order to explain the strict inclusion and exclusion criteria one has to use for constructing the dataset. A paucibacillary lesion may include very few bacilli easy to miss even by an experienced pathologist. This is why we excluded from the negative cases group the slides with morphological appearance highly suggestive of tuberculosis even if the ZN-stained slides did not reveal any bacilli no matter how thoroughly was the examination both at the moment of diagnosis and at reexamination. Additionally, in the B group, in the negative set of cases used for testing, cases with morphology similar to TB but with a clear diagnosis of diseases other than TB were included. This was conducted in order to avoid an involuntary bias created when the pathologist examines a ZN-stained slide that he/she is convinced that the diagnosis is not TB and obviously no bacilli may be present: “it does not look as TB, for sure no AFB are present; no careful scrutiny is needed”; “it looks as TB, maybe there are AFP present; and let’s look for them carefully”.

Xiong et al., describe reevaluation of the cases during the process of developing the algorithm. They reclassified seven cases initially labeled as negative [15].

Zaizen et al., have an interesting approach when constructing the testing group: the positive cases were those with proven mycobacteriosis either when the biopsy was performed or during follow-up; based on this perspective, AI-supported pathological diagnosis identified 11 positive cases versus 2 positive cases in classical pathological diagnosis, without AI support [19]. It is unusual for a pathologist to miss 9 cases from a total of 42 (12.5% sensitivity). The algorithm was able to identify 11 positive cases (2 cases identified as positive by human examiner and 9 more cases) and “missed” 5 cases. Due to the design of the testing process, these “missed” cases could be real negative ones at the moment of examination (if a patient is developing an illness in the future he or she is not mandatory presenting the microorganism months in advance) or, due to the scarcity and the not uniform distribution of the mycobacteria within the tissue it is possible that the tissue examined by algorithm did not contain bacilli in the moment of investigation.

Another important advantage of our dataset is represented by the number of the cases selected for annotation and the number of positive patches. We annotated 110 positive WSIs obtaining 263,000 positive patches. As it is shown in Table 8, this is the biggest and most diverse AI training dataset for mycobacteria to date. The number of negative cases is also important; at first glance, few negative WSIs are necessary for obtaining a large number of negative patches (one slide with 1 cm^2^ of tissue can be cut in more than 800,000 patches of 64 × 64 pixels). It is important, however, to have different types of tissue with different types of lesions to ensure a sufficient variability of the patches in both structure and color. The absolute number of negative WSIs of our training group is also the biggest, comparable with Pantanowitz et al., dataset but with several orders of magnitude higher than the others. The high number of ZN-stained slides is important due to the fact that it offers a higher diversity of images. ZN stain is a manual stain with high variability from lab to lab, being almost impossible to standardize. A “good” ZN stain is one that reveals mycobacteria from light pink to deep red or even purple rods on a light blue to dark blue background. In fact, its variability is so high that one technician cannot obtain two identical ZN stains on the same tissue block. This can be “a blessing in disguise” since the algorithm trained on a sufficiently large dataset (originating from a sufficiently numerous different WSIs), supplementary extended by augmentation techniques altering color, contrast, brightness, saturation, etc. will be able to properly recognize ZN-stained WSIs provided by labs worldwide.

In the five methods of automatic identification of mycobacteria in ZN-stained slides described in the literature, one study (Lo et al. [17]) does not evaluate WSI. Its validation is solely made on patches. Xiong et al. [15] present a completely automated method of diagnosis while Yang et al. [16], Pantanowitz et al. [18], and Zaizen et al. [19] developed AI-assisted diagnostic methods as a tool in the hands (and eyes) of pathologists. In Yang et al.’s method, the pathologist evaluates a score heatmap superposed on the WSI. In Pantanowitz et al.’s method, the pathologists evaluated a gallery of patches displayed in reverse order of the probability score in relation with WSI. Both methods allow for the pathologist to evaluate the suspicious areas in the context of the specific histopathological lesion. Zaizen et al., do not describe precisely how the pathologist uses the platform for diagnosis. Instead, they specify that each probably positive patch was examined by six pathologists.

Our method is an AI-assisted diagnostic method with a similar approach to Pantanowitz’s et al.’s design of the platform (analyzing a list of patches displayed in reverse order of the probability score). However, our solution employs a much larger dataset (about two orders of magnitude larger) and an active learning approach that further increases the performance metrics, especially for difficult cases (i.e., artifacts) or WSIs with peculiar staining.

Our algorithm obtained very good results compared with previous studies. Our testing method compared the AI-assisted diagnosis with the pathologist’s diagnosis either on slides (by microscopic examination) or WSIs. Our test set included 60 cases based on general recommendation for the minimal size required for digital pathology validation [33]. As expected, there is a definite improvement of AFB identification by pathologists when examining slides other than WSIs. It is known that pathologists are not very keen to change from conventional microscopy to remote WSI examination as a routine. The diagnostic concordance between WSI and slide examination varies from 63% to 100% in different studies [34].

Moreover, pathologist’s experience in examining WSIs affected the accuracy of finding AFB—the longer the period of exposure to WSIs, the better the pathologist’s results. The accuracy of the diagnosis when our algorithm was used was higher than the accuracy of every pathologist, even when slides were examined. The algorithm was able to pick more bacilli than the human examiner alone, thus almost eliminating the false negatives. When examining slides, pathologists missed a total of 47 cases of TB (false negatives), in average almost 6 cases per person. Our algorithm helped pathologists improve Mycobacterium identification on WSIs, but the results were also better with AI-assisted evaluation than those where pathologists examined slides by microscope. In real life, a pathologist examining a slide may identify lesions suspicious of TB—epithelioid granulomas with giant cells and/or coagulative necrosis with reticuline preservation (caseum). When one suspect TB, he/she will ask for a ZN special stain in order to identify bacilli. AFB presence confirms the diagnosis of TB without biunivocal relation (i.e., AFB absence does not exclude TB diagnosis). In other words, when a pathologist fails to identify AFB, he/she will not necessarily miss TB but the positive diagnosis that will finally be obtained in most of the cases will be obtained with supplementary efforts (several costly techniques) and with some delays in significant cases. Altogether, both the patients and the medical system will benefit from implementation of such an algorithm in routine pathology.

Another issue for discussion is the debate about what metric should be preferred: specificity or sensitivity? A diagnostic method is preferable to be specific while a screening test is better to be more sensitive. We decided to use a higher specificity (fewer false positive cases) with the risk of missing some positive cases (false negative). The algorithm selects patches that are more probable to contain AFB and shows them to the pathologist. If the algorithm is picks up too many structures, the pathologist will be forced to look to a myriad of artifacts and he/she will lose a lot of time sorting through them.; In the end, it is more profitable to examine the slide without AI support.

Last but not least, when discussing our algorithm capabilities in comparison with human results, we should not forget that our team of pathologists are familiar with ZN stains and AFB identification on slides; we expect that a pathologist not used to examining ZN-stained slides would have poorer results with more numerous false negatives, especially in paucibacillary lesions.

When looking at the errors in analyzing both WSIs and slides, there are huge differences between qualified pathologists and residents. The residents were in their final year of residency and are very good and hard-working people. However, we showed that no exposure to WSIs prior to this test poorly influenced the outcome.

We have a closer look at the cases with the most numerous errors in interpretation. One negative case had five errors with eight examiners and four errors from six qualified pathologists (cat scratch disease—suppurative necrotizing granulomatous lymphadenitis). Some structures looked like AFB, but the overall quality of the stain was poor (slightly pink–pale red blood cells). In some areas, structures could be mistaken as AFB but the suspicious structures were not clear-cut bacillar structures (Figure 13 and Figure 14).

A case of tuberculosis in striated muscle had four errors from eight examiners. There were many fragments of tissue and almost 5 cm^2^ of tissue with very few bacilli, which were easily missed by examiners (Figure 15).

The case with most errors in interpretation was a tuberculous epithelioid granulomatous lymphadenitis with extensive caseation with very few bacilli present in ZN stain—one to four AFB present in each section. Due to the minute dimensions of Mycobacterium tuberculosis (one micron thick), a bacillus will be completely enclosed in one section of tissue and serial sections reveal different bacilli. The slide examined in this test included two sections of tissue with very few bacilli, one in one section (Figure 16) and two on the other section. Considering the paucity of the bacilli, it is no wonder that the examiners missed them on WSIs. Interestingly, this was the case the algorithm was not able to identify bacilli. For this case, the algorithm identified 3 patches with positive scores over 0.7 and 145 patches with positive scores between 0.5 and 0.69. None of them presented convincing AFBs.

In fact, in order to avoid the examiner being biased by the overall picture of the lesion, the testing set was designed to include lesions with similar appearance to tuberculosis such as granulomatous inflammation, most of them with necrosis. The cases with reactive lymphadenitis, unspecific inflammation, or malignancies were not erroneously evaluated by any examiner.

The most impressive benefit of using the AI-assisted algorithm for AFB identification is saving time. AI-assisted evaluation was 2.84 times faster than human evaluation. We have to be aware that the pathologists involved in our clinical test had impressive experience in diagnosing tuberculosis and analyzing ZN stains. Our department has expertise in infectious diseases diagnosis. ZN stain is routine for lymph nodes and bronchial biopsies and, moreover, the pathologists were recently exposed to numerous positive and negative ZN-stained WSIs during the annotation period. A “regular” pathologist likely does not have the same level of exposure, so the time required for a thorough examination of a ZN-stained slide is usually much longer. We can estimate that our algorithm saves at least one-third of the pathologist’s time that can be spent on other more complex tasks.

Moreover, considering the inherent bias induced by the level of expertise of our team of pathologists, the results of our model argue in favor of an overall increase in the quality of AI-assisted diagnosis. In other words, if the model was able to reach the best performance of one of our most experienced pathologists (identifying convincing positive patches in all but one cases), for a less experienced pathologist the algorithm will certainly improve their performance. It is true that the final labeling of the status of the patch (positive versus negative) is established by the pathologist. The fact that the model is identifying highly suggestive areas helps the human examiner to make a final decision.

Situations when AI algorithms performed better than pathologists were reported during clinical testing for automatic identification of prostate cancer. One case previously missed by pathologists was suggested as malignant by the algorithm and confirmed as such by experts [35]. Additionally, algorithms for Mycobacterium tuberculosis detection identified positive cases with subsequent expert’s confirmation [15,19].

Our algorithm is able to identify bacilli even in cases with a very low density of AFB and in cases that were missed by pathologists, even when considering experienced pathologists (AI-assisted diagnosis based on our method has a specificity 100% and sensibility 95.65%). The impact of this achievement is significant. Our automatic method being used to assist pathologists in identifying AFB is saving time and money that is otherwise required by other investigations. Therefore, it shortens the interval between the biopsy and the start of the treatment with major benefits, both for the patient (better results and faster improvement of health) and for society (faster decrease in the patient’s infectiousness, diminishing the medical costs for expensive investigations or, longer treatments required for old lesions, diminishing the social security costs by fewer days of medical leave, etc.).

There are many limitations for our technique. The main important limitations concern the dimensions and diversity of the dataset and our method of clinical testing.

Our dataset is the largest and most diverse of the datasets for mycobacteria presented in the literature. It is also the most “correct” one, due to our method selecting cases. Unfortunately, it is not a “perfect” dataset; to reach this goal, the dataset should include all the positive slides from all over the world. This is virtually impossible. We applied several techniques of augmentation to minimize this drawback, but we are aware of this impossible to overcome drawback.

Our method of clinical testing is also flawed because of the simple fact that the team of researchers who developed the algorithm also validated it. This forced manner of designing the test of the algorithm thus biases the validation of all AI-models developed in medicine. We tried to diminish this risk by separating the people who designed and selected the test group of cases from the people who actually performed the test. Our most experienced pathologist tried to further minimize the risk by including in the test group positive paucibacillary cases and negative cases with similar microscopic appearance to TB (see the discussions above). We are aware that the bias is not completely overcome due to the simple fact that the pathologists belong to the same school of pathology with similar methods of evaluation and routines. The only answer for this limitation is for independent validation to be performed by pathologists from completely different institutions and from as many countries as possible, ideally on international cohorts of patients. Overcoming this problem represents the key towards clinical implementation of the algorithm [36].

## 5. Conclusions

We developed a model for AI-assisted detection of AFB on WSIs that is able to identify bacilli with an accuracy of 98.33%, sensitivity of 95.65%, and specificity of 100%. The results were better than or, for one case, similar to those of a team of pathologists of variable expertise when examining slides and WSIs, thus reducing human error form fatigue and loss of focus. By using our algorithm, pathologists saved at least one-third of the total examining time.

We intend to annotate the positive WSIs used for clinical testing and retrain our algorithm with the resulting supplementary patches, thus making use of our active learning setup. The new product iteration will be further tested in different hospitals to test the robustness of the algorithm when exposed with different types of ZN stains and to diminish the inherent subjectivity of the validation.

## Figures and Tables

**Figure 1 diagnostics-12-01484-f001:**
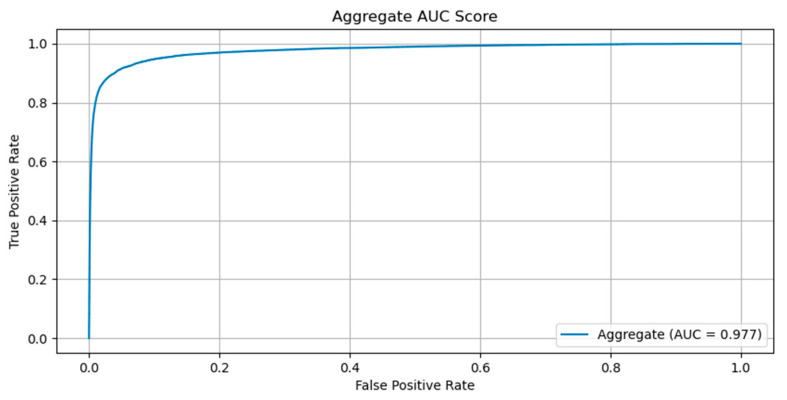
ROC curve obtained by the model on the validation set.

**Figure 2 diagnostics-12-01484-f002:**
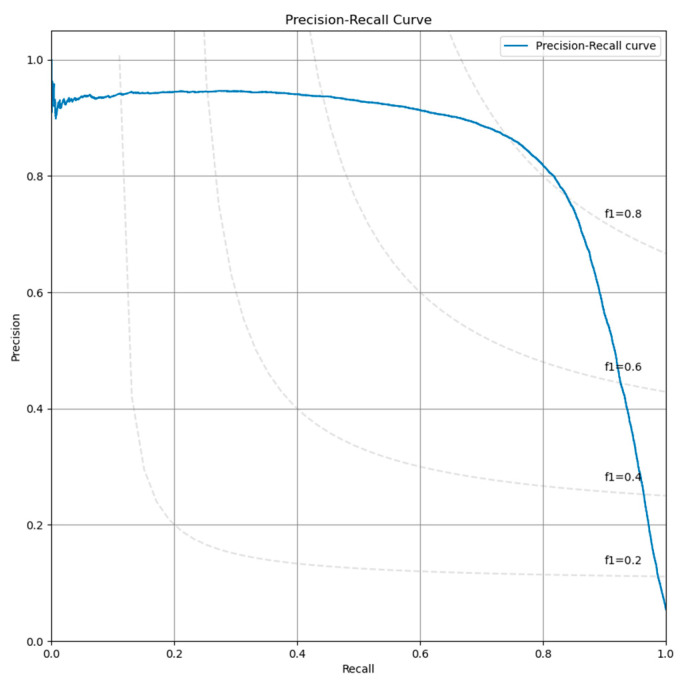
Precision-recall curve obtained by the model on the validation set.

**Figure 3 diagnostics-12-01484-f003:**
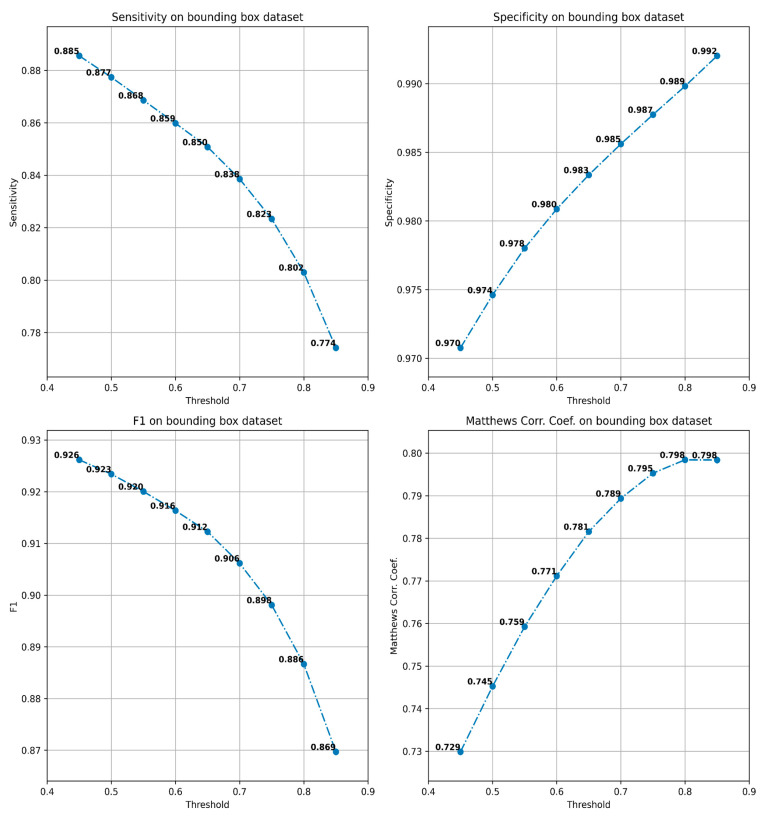
Sensitivity, specificity, F1-score, and MCC of the model computed for various thresholds.

**Figure 4 diagnostics-12-01484-f004:**
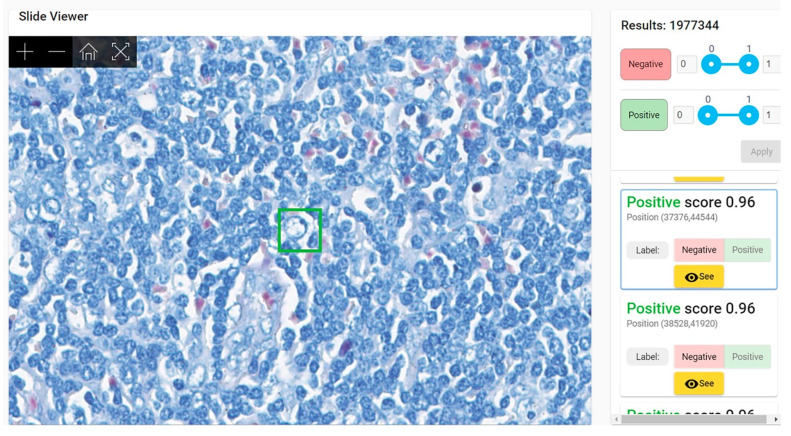
Patch of 64 × 64 pixels (in green) with a positive score (probability of similarity with positive dataset used for training) of 0.96 due to the presence in the upper left margin of the green square of a red blood cell. Lymph node with toxoplasmosis ZN × 400.

**Figure 5 diagnostics-12-01484-f005:**
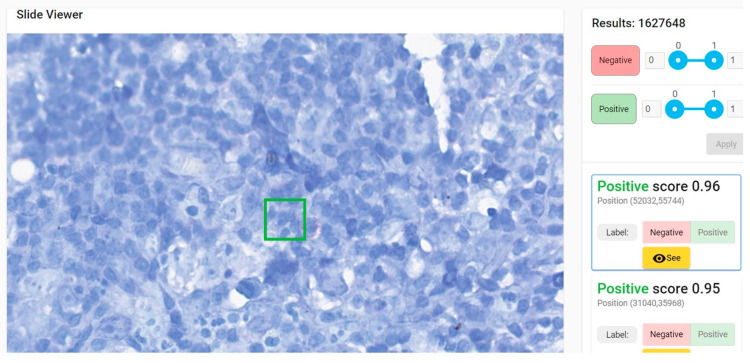
Patch of 64 × 64 pixels (in green) with a positive score of 0.96 due to the presence in the inferior right margin of the green square of several purple mast cell granules with linear arrangement mimicking an acid-fast bacillus. Hodgkin’s lymphoma, nodular sclerosis variant. ZN × 400.

**Figure 6 diagnostics-12-01484-f006:**
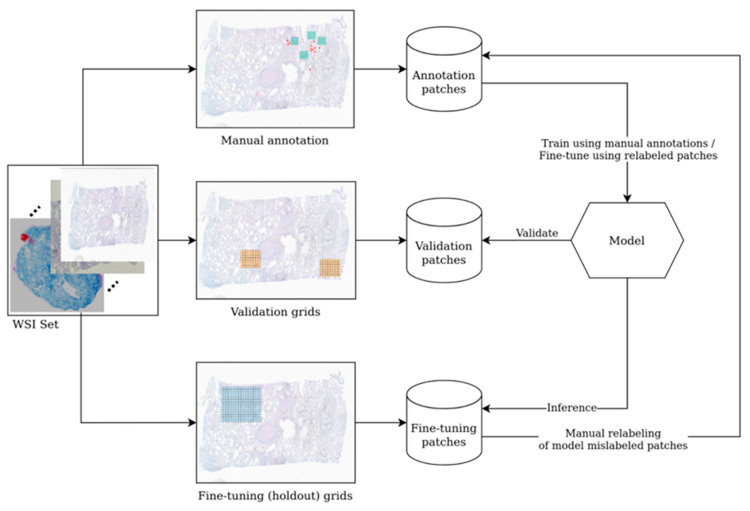
Active learning process for iteratively improving the model performance.

**Figure 7 diagnostics-12-01484-f007:**
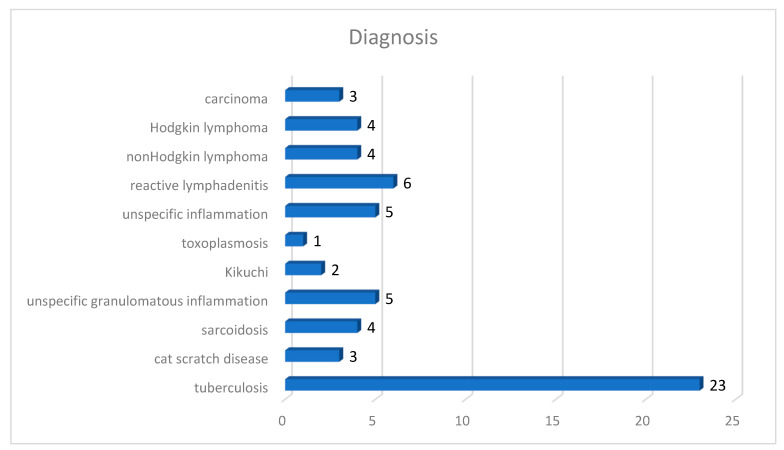
Repartition of the test group according to diagnosis.

**Figure 8 diagnostics-12-01484-f008:**
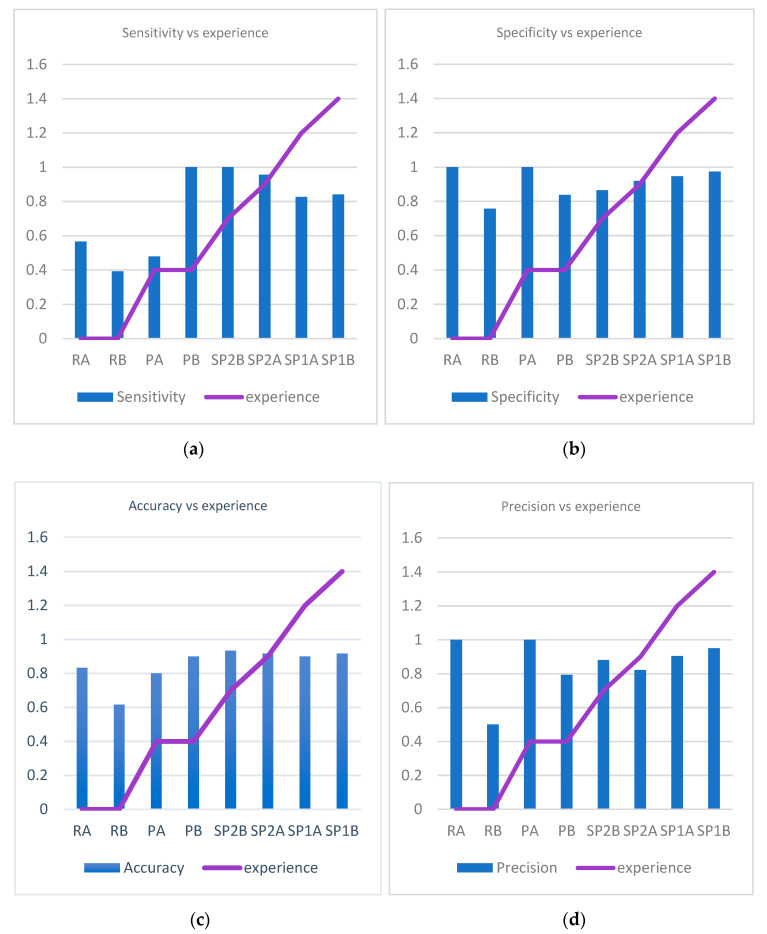
Variation of sensitivity, specificity, accuracy, and precision in correlation to pathologists’ experience. (**a**) Variation of sensitivity in correlation to pathologists’ experience. (**b**) Variation of specificity in correlation to pathologists’ experience. (**c**) Variation of accuracy in correlation to pathologists’ experience. (**d**) Variation of precision in correlation to pathologists’ experience.

**Figure 9 diagnostics-12-01484-f009:**
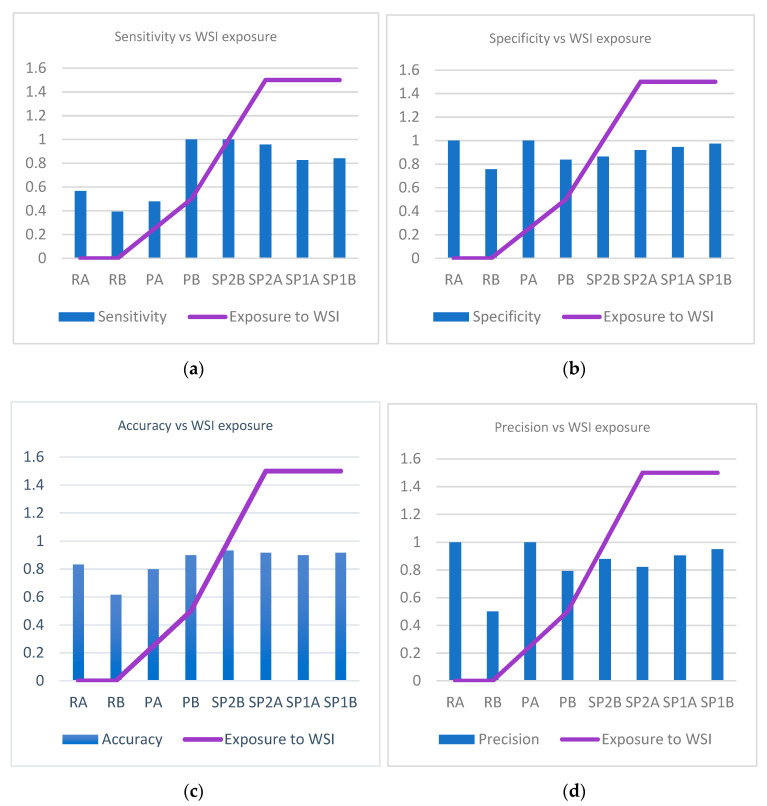
Variation of sensitivity, specificity, accuracy and precision in correlation to pathologists’ experience of examining whole slide images. (**a**) Variation of sensitivity in correlation to pathologists’ experience of examining WSIs. (**b**) Variation of specificity in correlation to pathologists’ experience of examining WSIs. (**c**) Variation of accuracy in correlation to pathologists’ experience of examining WSIs. (**d**) Variation of precision in correlation to pathologists’ experience of examining WSIs.

**Figure 10 diagnostics-12-01484-f010:**
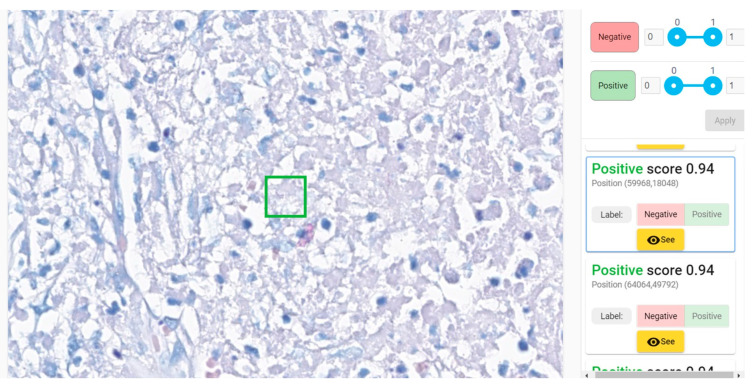
Paucibacillary lesion identified as positive by 5 of 8 pathologists in 1–32 min (medium of 13.75 min); the time of AI-assisted examination was 15 s (the convincing positive patch—the green square—was the second one).

**Figure 11 diagnostics-12-01484-f011:**
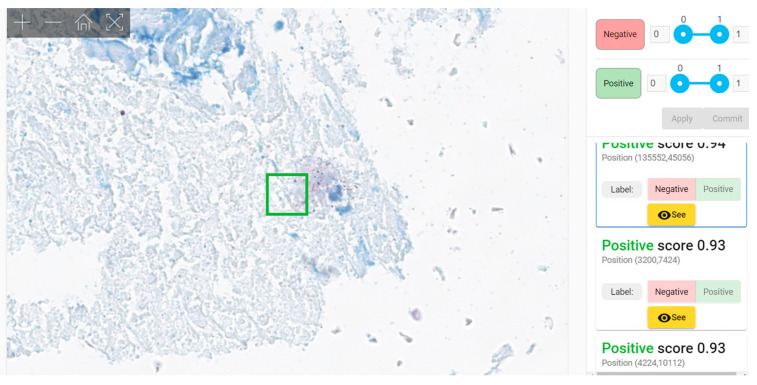
Paucibacillary lesion identified as positive by 6 of 8 pathologists in 1–33 min (medium of 12.25 min); the time of AI-assisted examination was 9 s (first patch—green square—was convincingly positive).

**Figure 12 diagnostics-12-01484-f012:**
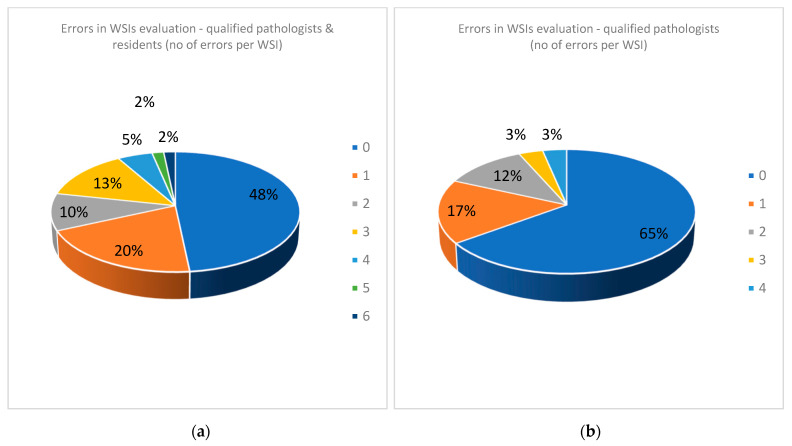
(**a**) Errors in WSIs evaluation for all the team (qualified pathologists and residents). (**b**) Errors in WSIs evaluation for qualified pathologists.

**Figure 13 diagnostics-12-01484-f013:**
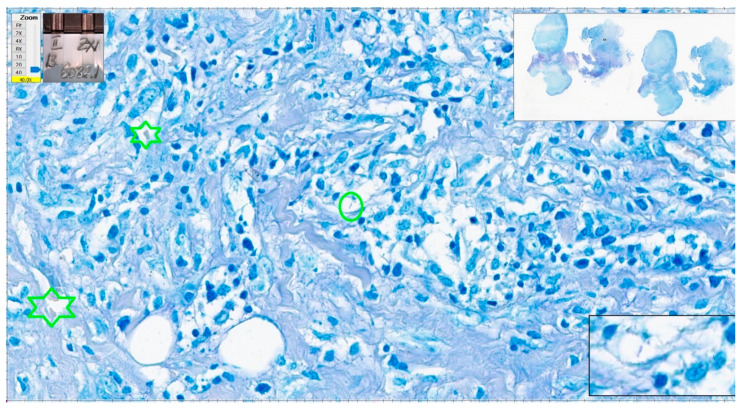
Cat scratch disease. Centrally, one structure reminiscent of AFB but pale blue in color (green oval); however, the color of red blood cells is not appropriate (green stars). Paler than regular in a good Ziehl–Neelsen stain. ZN × 400 as offered by Aperio ImageScope platform; WSI scanned with Aperio GT450, 40× magnification.

**Figure 14 diagnostics-12-01484-f014:**
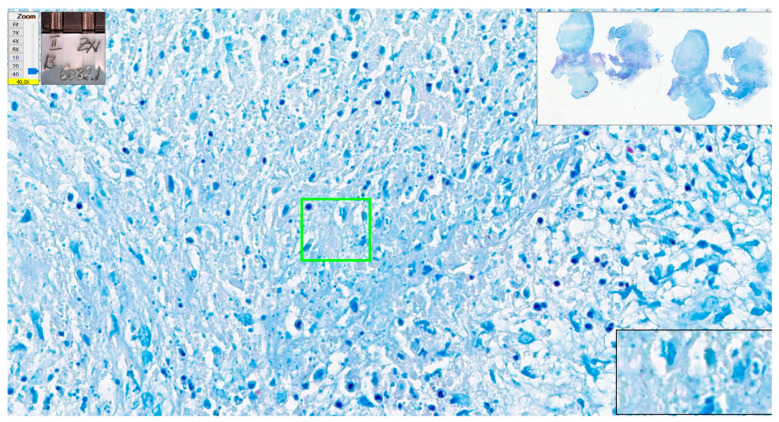
Cat scratch disease. Centrally, several structures look like AFB but pale blue in color (green rectangular area); however, enhancement of the image—black contour window in the lower right corner of the picture (digital magnification offered by Aperio ImageScope software)—shows improper format of the pink structures. ZN × 400 as offered by Aperio ImageScope platform; WSI scanned with Aperio GT450, 40× magnification.

**Figure 15 diagnostics-12-01484-f015:**
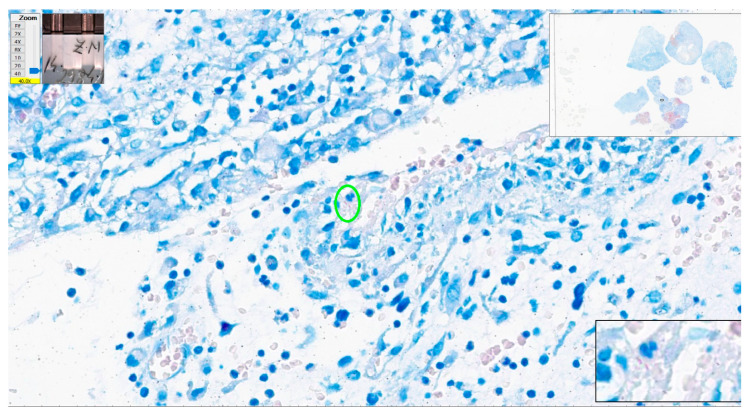
Tuberculosis. Centrally, two AFBs present (green oval); please note the good quality of the Ziehl–Neelsen stain certified by the pink color of red blood cells. ZN × 400 as offered by Aperio ImageScope platform; WSI scanned with Aperio GT450, 40× magnification.

**Figure 16 diagnostics-12-01484-f016:**
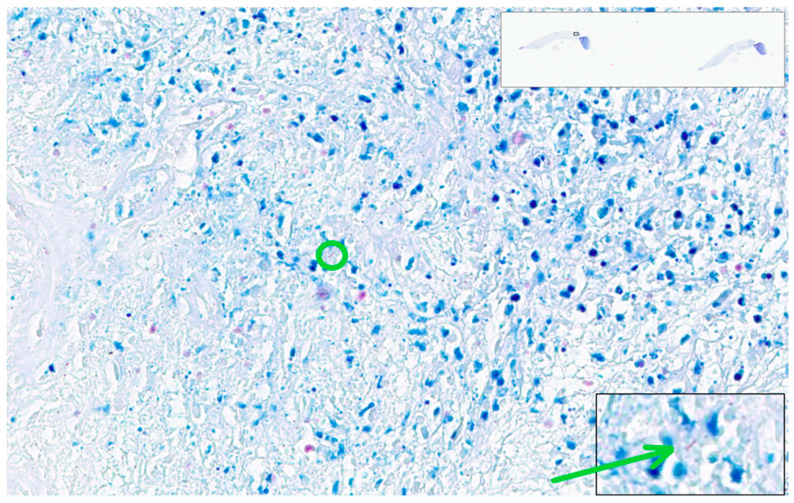
Tuberculous epithelioid granulomatous lymphadenitis with extensive caseation. One AFB is present within the center (green circle). Higher resolution is in the right inferior rectangular area (detail: green arrow). ZN × 400 as offered by Aperio ImageScope platform; WSI scanned with Aperio GT450, 40× magnification.

**Table 1 diagnostics-12-01484-t001:** Studies of automated detection of AFB on ZN stains on smears.

Studies on Smears	Year	Precision	Sensitivity	Specificity
Ayas et al. [6]	2014	N/A *	89.34	96.97
delCarpio et al. [4]	2019	N/A	93.67	89.23
Costa et al. [5]	2008	N/A	76.65	N/A
Costa Filho et al. [7]	2015	N/A	96.8	N/A
El-Melegy et al. [8]	2019	82.6	98.3	N/A
Khutlang et al. [9]	2010	N/A	97.77	N/A
Kuok et al. [10]	2019	N/A	98.06	91.65
Law et al. [11]	2018	N/A	70.4	76.6
Panicker et al. [12]	2018	78.40	97.13	N/A
Vaid et al. [13]	2020	88.4	92.1	N/A
Veropoulos et al. [3]	1999	N/A	94.1	97.4
Zhai et al. [14]	2010	N/A	89.34	96.97

* N/A—not available.

**Table 2 diagnostics-12-01484-t002:** Structure of the study group.

Group	Positive	Negative	Total
A (training)	110	400	510
B (testing)	23	37	60
Total	133	437	570

**Table 3 diagnostics-12-01484-t003:** Statistical measures of the pathologists’ performance on whole slide images.

	Senior Pathologist 1A	Senior Pathologist 1B	Senior Pathologist 2A	Senior Pathologist 2B	Pathologist A	Pathologist B	Resident A	Resident B	Our model
Sensitivity	0.8261	0.8261	0.9565	1	0.4782	1	0.5652	0.3913	0.9565
Specificity	0.9459	0.9730	0.9189	0.8648	1	0.8378	1	0.7567	1
Precision	0.9048	0.9500	0.8800	0.8214	1	0.7931	1	0.5000	1
Negative predictive value	0.8974	0.9000	0.9714	1	0.7551	1	0.7872	0.6667	0.9737
False negative rate	0.1739	0.1739	0.0435	0	0.5217	0	0.4348	0.6087	0.0435
False positive rate	0.0541	0.0270	0.0811	0.1351	0	0.1622	0	0.2432	0
Accuracy	0.9000	0.9167	0.9333	0.9167	0.8000	0.9000	0.8333	0.6167	0.9833
F1	0.8636	0.8837	0.9167	0.9019	0.6471	0.8846	0.7222	0.4390	0.9778
Exposure to WSI (years)	1.5	1.5	1.5	1	0.25	0.5	0	0	-
Experience (decades)	1.2	1.4	0.7	0.9	0.4	0.4	0	0	-

**Table 4 diagnostics-12-01484-t004:** Statistical measures of the pathologists’ performance on glass slides.

	Senior Pathologist 1A	Senior Pathologist 1B	Senior Pathologist 2A	Senior Pathologist 2B	Pathologist A	Pathologist B	Resident A	Resident B	Our model
Sensitivity	0.9130	0.8695	0.9130	0.9565	0.5217	0.9565	0.4782	0.3478	0.9565
Specificity	1	1	0.9729	1	0.9729	0.9189	1	0.9459	1
Precision	1	1	0.9545	1	0.9230	0.88	1	0.8	1
Negative predictive value	0.9487	0.925	0.9473	0.9736	0.7659	0.9714	0.7551	0.7	0.9737
False negative rate	0.0869	0.1304	0.0869	0.0434	0.4782	0.0435	0.5217	0.6521	0.0435
False positive rate	0	0	0.0270	0	0.0270	0.0811	0	0.0540	0
Accuracy	0.9667	0.95	0.95	0.9833	0.8	0.9333	0.8	0.7166	0.9833
F1	0.9545	0.9302	0.9333	0.9778	0.6667	0.9167	0.6471	0.4848	0.9778
Experience (decades)	1.2	1.4	0.7	0.9	0.4	0.4	0	0	-

**Table 5 diagnostics-12-01484-t005:** Time used by pathologists when examining whole slide images.

Time of Examination (min)	Senior Pathologist 1A	Senior Pathologist 1B	Senior Pathologist 2A	Senior Pathologist 2B	Pathologist A	Pathologist B	Resident A	Resident B
All WSIs	1–45	1–35	0.08–20	0.16–26	0.5–80	0.33–35	9.5–35	0.1–28
Average all WSIs	17.07	12.38	5.48	5.78	14.04	13.58	14.95	8.16
True positive	1–18	1–18	0.08–15	0.16–8	0.5–25	0.33–21	0.5–32	0.1–13
Average true positive	7.26	6.54	4.28	2.47	5.59	7.64	12.54	3.51
False positive	9–14	27	4–15	0.16–8	-	4–30	-	3–21
Average false positive	11.50	27.00	11.33	4.43	-	10.67	-	7.33
True negative	8–45	2–35	1–20	1–26	0.5–80	7–35	3–35	2–20
Average true negative	21.37	11.56	5.88	8.38	12.38	18.55	14.14	7.25
False negative	15-36	5-32	1	-	2-52	-	7-29	5-28
Average false negative	28.75	21.50	1.00	-	26.92	-	21.10	13.50

**Table 6 diagnostics-12-01484-t006:** Time used by pathologists when examining slides by microscope.

Time of Examination (min)	Senior Pathologist 1A	Senior Pathologist 1B	Senior Pathologist 2A	Senior Pathologist 2B	Pathologist A	Pathologist B	Resident A	Resident B
All slides	1–20	1–49	0.05–10	0.16–26	0.5–38	0.33–16	0.5–22	0.5–19
Average all slides	6.13	7.08	2.84	4.25	5.44	5.06	6.15	5.09
True positive	1–19	1–32	0.05–10	0.16–8	0.5–10	0.16–15	0.5–18	0.5–11
Average true positive	3.71	11.2	3.21	3.42	3.04	5.46	0.92	2.75
False positive			8		0.5	3-16		1-3
Average false positive	-	-	8	-	0.5	11.33	-	2
True negative	1–20	1–16	0.05–10	1–26	0.5–38	0.33–11	0.5–12	1–11
Average true negative	7.46	3.45	2.52	4.76	4.77	3.98	4.42	4.1
False negative	5–9	2–49	1–4	4	1.5–25	13	1–22	2–19
Average false negative	7	24.33	2.5	4	10.68	13	8.15	9.06

**Table 7 diagnostics-12-01484-t007:** Errors in WSIs evaluation for qualified pathologists and residents.

Qualified Pathologists and Residents (8 Persons × 60 WSI)				
		Negative Cases	Positive Cases	Total
No of errors per WSI	0	23	6	29
	1	9	3	12
	2	1	5	6
	3	3	5	8
	4	0	3	3
	5	1	0	1
	6	0	1	1
Cases with errors (of 60 WSIs)		14	17	31
%		37.84%	73.91%	51.67%
No of errors (of 480 examinations)		25	46	71
%		8.45%	25.00%	14.79%

**Table 8 diagnostics-12-01484-t008:** Studies of automated detection of AFB on ZN stains on tissue.

Studies on Tissue	Year	Training Set Positive WSIs	Training Set Total WSIs	Patches Positive	Patches Negative	Patches (Pixels)	Test Set	Accuracy %	Sensitivity (Recall) %	Specificity %
Xiong et al. [15]	2018	30	45	96,530	2,510,307	32 × 32	201 WSIs	90.55 *	97.94	83.65
Yang et al. [16]	2020	6	33	18,246	18,246	256 × 256	134 WSIs	87 *	87.13	87.62
Lo et al. [17]	2020	9	9	613	1202	20 × 20	patches	95.30	93.5	96.3
Pantanowitz et al. [18]	2021	47	418	5678	1,111,918	32 × 32	138 WSIs	84.6	64.8	95.1
Zaizen et al. [19]	2022	2	42	506	N/A	N/A	42 WSIs	N/A	86	100
Our study	**2022**	110	510	**263,000**	**700,000,000**	**64 × 64**	**60 WSIs**	**98.33**	**95.65**	**100**

* calculated based on the data provided in the paper.

## Data Availability

Not applicable.

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
