# Peer review of "A New Artificial Intelligence-Based Method for Identifying Mycobacterium Tuberculosis in Ziehl–Neelsen Stain on Tissue"

_diagnostics, 2022, doi:10.3390/diagnostics12061484_

Round 1

Reviewer 1 Report

In their work, the authors proposed a CNN-based approach to aid in identification of slides that are positive for AFB. The CNN model was trained using high-resolution images of ZN-stained tissues digitized on the Aperio GT450. This study is in line with past research in terms of the overall pipeline design; it includes a classification CNN and a variety of tissues stained with the ZN-stain. The AFB-positive slides were complemented by the AFB-negative slides from multiple organs and diseases which in some cases display patterns resembling a positive ZN stain in AFB.  Collecting challenging cases/slides indicates that the study was well designed and the team includes experienced pathologists who can identify diagnostically difficult areas for the computer scientists to properly train and evaluate the CNN.  The study is valuable because it highlights issues in digital pathology pertaining to the development of robust AI tools for the identification of AFB-positive and -negative slides.  However, there are areas of concern regarding methodology and manuscript organization which need to be addressed.

Major concerns:

-          What is the point of using H&E-stained slides in this study? The AFB microorganisms are not visible in slides stained with H&E.

-          Description of Zaya is completely missing. It appears that Zaya is either a company or algorithm name, or both. Since no links o references are provided, a detailed description of Zaya (assuming it is an algorithm) is required.

-          It is unclear whether the performance metrics reported in the manuscript relate to the patch or slide based classifications.  More unclear is how whole slides are classified as AFB-positive and AFB-negative. A detailed description of this crucial step needs to be provided - ideally in a separate section.

-          Explain clearly what is algorithm-aided WSI evaluation? Is this a completely computerized and automated classification of WSIs, or it is a system that highlights areas containing AFBs for the pathologists to make the decision about the AFB positivity for the whole slide?   

-          The manuscript is overwhelmed with figures and tables. The authors can reduce the number of figures by moving Figs 1-6 to the supplement. Same concern applies to Tables 1-5. The content of other figures and tables should also be reduced. For instance, by removing GUI/interface windows and other content that is scientifically irrelevant.

-          If the authors are determined to keep GUIs with Zaya logo, they should explain every single component and info displayed in the GUI and discuss how it helps the pathologist in the diagnosis of ZN-stained slides. The explanation can be included in the figure caption or in a separate section/paragraph.  If the scientific value of the GUI and displayed components are marginal in the context of this study, they should be removed.

-          Introduction misses the description of state of the art which was instead placed in Discussion.  The authors are asked to move relevant paragraphs to Introduction. The introduction should be summarized by a statement about novelty and main contribution of the study.

-          In the rewritten Introduction, the authors should include missing literature:

-          https://www.sciencedirect.com/science/article/abs/pii/S0895611120300550

-          https://dl.acm.org/doi/abs/10.1007/978-3-030-31332-6_24

-          https://pubmed.ncbi.nlm.nih.gov/29357378/

-          https://journals.plos.org/plosone/article?id=10.1371/journal.pone.0212094

and mention these papers in Table 14.  In Discussion, they should discuss findings from their study in the context of those listed in revised Table 14. Particularly important is the discussion about what their algorithm did better than the state-f-the-art and why it was better.

Minor:

-          Please replace jargon such as “Time consumed by pathologists “ with a syntax that is more adequate to appear in a scientific publication.

-          The current title is awkward. Consider: A New Artificial Intelligence-based Method for Automatic Identification of Mycobacterium Tuberculosis in Ziehl Nielsen-stained slides, or provide one that has a better syntax.   

-          Instead of “golden standard” , should be: “gold standard”

Reviewer 2 Report

Please find the specific comments as a separate .pdf file.

Reviewer 3 Report

The paper proposes an automated (AI-based) method of identification of mycobacteria. The paper contains some standard approaches, and the novelty is vague. Also, some paragraphs are unreadable and needs to be written including the abstract

The abstract: the abstract needs to be revised. The statement “(AI-based) method of identification of myco-bacteria.” Is repeated twice. Some phrases are unreadable. The statement “Our architecture showed 0.977 on ROC curve” it should be area under the ROC curve. There are several papers that proposed automatic systems for the diagnosis of tuberculosis using artificial intelligence techniques what is the novelty of the paper? Could you please mention the novelty in the abstract?

Introduction

Abbreviations should appear the first time they appear like TB.

Many parts of the introductions are unreadable.

Mycobacteria is sometimes written in capital letters and sometimes small.

I finished the introduction, and I cannot find the novelty and contribution. Tuberculosis diagnosis using AI is not a new area of research plenty of work has been conducted. What are your novelty and contribution?

I cannot see a related work and literature review section. Any scientific paper should include related work either in the introduction or a dedicated section.

Could you please add some related studies and mention any of their limitations that motivated you to propose a new approach for tuberculosis diagnosis. Please discuss the advantages and limitations of these techniques.

Material and Methods:

What is SZ?

Please add a figure describing the steps of your proposed method.

Again, I cannot find any contribution or novelty.

Experimental Results

1.     Please define the performance metrics used to evaluate your proposed method in a dedicated section and write their formula.

2.     Since the dataset is unbalanced please add the F1 measure and Mathew correlation coefficient to the results section

3.     Please mention the limitations of your technique.

The conclusion please mention your future work.

Round 2

Reviewer 1 Report

The manuscript has been significantly revised and improved. However, one aspect of this work remains hidden on purpose or poorly described, that is, how the proposed pipeline, by itself, identifies AFB-positive and AFB-negative slides.

It seems that there is no mechanism built into the proposed pipeline to classify whole slides. Here, I mean a mechanism that can visit EACH patch on the slide, assess it for the possibility of containing AFB and use information from ALL the assessed patches to finally classify the slide. This type of slide classification could then be compared to the gold-standard binary pathologist classification (AFB-positive slide or AFB-negative) to measure pipeline's performance. 

As it stands now, described experiments suggest that the pipeline's performance was measured using patches from pre-SELECTED regions on slides instead of ALL patches from whole slides, thereby confusing readers about how the wholes slides were classified and how the pipeline’s whole slides classification performance was evaluated.

Since the authors claim that their pipeline can classify whole slides (Table 8), they need to clearly describe how this was done.  The reviewer suggests that a full description of the whole slide classification approach is placed in the Methods, and a short mention in the Abstract as well.

As the Discussion is mostly focused on discussion patch-based classification performances, it should be augmented by highlighting main differences between the proposed slide classification approach and those listed in Table 8.

Author Response

Dear Sir/ Madam,

Thank you for your valuable comments. Indeed, a more precise description of the method of slide classification in correlation with the work of previous studies is necessary.

We added in the Material and Methods the modality for WSI classification; in the introduction, in 1.2. Literature review section, we added short description of the WSI classification for each author. In the Discussion chapter we added a paragraph comparing the 5 papers from the literature with our method. We also added a short sentence in the abstract.

Sincerely yours,

Sabina Zurac et al

Reviewer 3 Report

The authors have addressed my comments but take care that there is an empty table below Table 7. 

Author Response

Dear Sir/ Madam,

Thank you for your valuable comments, you helped us to significantly improve the quality of our manuscript.

Sincerely yours,

Sabina Zurac et al